# Insight into the Mechanism of Lysogeny Control of phiCDKH01 Bacteriophage Infecting Clinical Isolate of *Clostridioides difficile*

**DOI:** 10.3390/ijms25115662

**Published:** 2024-05-23

**Authors:** Adam Iwanicki, Małgorzata Roskwitalska, Natalia Frankowska, Dorota Wultańska, Monika Kabała, Hanna Pituch, Michał Obuchowski, Krzysztof Hinc

**Affiliations:** 1Division of Molecular Bacteriology, Medical University of Gdańsk, 80-211 Gdańsk, Poland; adam.iwanicki@gumed.edu.pl (A.I.); m.roskwitalska.630@studms.ug.edu.pl (M.R.); n.frankowska.794@studms.ug.edu.pl (N.F.); michal.obuchowski@biotech.ug.edu.pl (M.O.); 2Intercollegiate Faculty of Biotechnology, University of Gdańsk, 80-307 Gdańsk, Poland; 3Department of Medical Microbiology, Medical University of Warsaw, 02-004 Warsaw, Poland; dorota.wultanska@wum.edu.pl (D.W.); hanna.pituch@wum.edu.pl (H.P.); 4Department of Medical Microbiology, Medical University of Silesia, 40-055 Katowice, Poland; mkabala@sum.edu.pl

**Keywords:** bacteriophage, *Clostridioides difficile*, lysogeny control

## Abstract

*Clostridioides difficile* is a causative agent of antibiotic-associated diarrhea as well as pseudomembranous colitis. So far, all known bacteriophages infecting these bacteria are temperate, which means that instead of prompt lysis of host cells, they can integrate into the host genome or replicate episomally. While *C. difficile* phages are capable of spontaneous induction and entering the lytic pathway, very little is known about the regulation of their maintenance in the state of lysogeny. In this study, we investigated the properties of a putative major repressor of the recently characterized *C. difficile* phiCDKH01 bacteriophage. A candidate protein belongs to the XRE family and controls the transcription of genes encoding putative phage antirepressors, known to be involved in the regulation of lytic development. Hence, the putative major phage repressor is likely to be responsible for maintenance of the lysogeny.

## 1. Introduction

*Clostridioides difficile* is a Gram-positive anaerobic bacterium capable of endospore production. It is commonly known as a causative agent of diarrhea associated with antibiotic therapy [1]. The onset of *C. difficile* infection (CDI) is strictly related to the disruption of the intestinal microflora caused by antibiotics [2,3]. The extreme resistance of *C. difficile* endospores to various physical, chemical, and biological factors enables their persistence in the environment, rendering this bacterium an important strain responsible for nosocomial infections. The histological manifestation of CDI is the damage of colon epithelial cells associated with acute inflammation and formation of pseudomembranes. The major virulence factors responsible for such symptoms are toxins (TcdA, TcdB, and binary toxins) produced by *C. diffile* following colonization of the colon [4].

As in the case of every bacterial species, *C. difficle* can be infected with its specific bacteriophages (reviewed in [5]). In this case, all so far characterized phages infecting these bacteria are temperate and belong to the class of *Caudoviricetes*. This class consists of tailed phages with an icosahedral head, dsDNA, and a variable tail length, formerly assigned to the order of *Caudovirales* (*Myoviridae*, *Siphoviridae*, *Podoviridae*) [6]. Despite belonging to a single taxonomic class, processes leading to lysogeny are different for individual phages. While the most common way requires integration of the viral DNA with the host genome [7,8], several *C. difficle* phages, upon onset of lysogenisation, are maintained episomally in the form of an independent plasmid. Examples of such phages are phiCD38-2 or phiCD211/phiCDIF1296T [9,10,11]. It is worth mentioning that infection of the host with some phages can lead to significant changes in its physiology, including an increase in the virulence of lysogenic strains. The phiSemix9B1 phage was the first to be shown to harbor the functional binary toxin gene [12].

Despite the temperate nature of *C. difficile* phages, they are capable of either spontaneous [13,14] or induced excision of the prophage and initiation of the lytic cycle. Cellular stressors leading to DNA damage are the known factors inducing prophages [15]. The most frequently used agents are UV [16], Mitomycin C [17], and Norfloxacin [18], each leading to the production of high phage titers obtained with lysogenic strains of *C. difficile*. The key question concerning keeping and breaking the lysogeny state is: what is the mechanism/mechanisms responsible for these phenomena? The prototypical temperate phage, the bacteriophage lambda, encodes a pair of repressors: CI, responsible for keeping the lysogeny state and Cro, counteracting lysogenic development of the phage [19]. In the case of *C. difficle* phages, the lysogeny/lysis decision has not been investigated in detail. While homology searches enabled the identification of genes encoding both repressors in the genomes of different *C. difficile* phages, the overall situation seems much more complicated. Genomes of *C. difficile* phages are modular, so different regions encode proteins with similar functions; nevertheless, the genes of CI and/or Cro homologs are not always present in them. Moreover, in the case of some *C. difficle* phages, numerous putative repressor and antirepressor genes can be found in their genomes, and the functions of these proteins mostly remain either speculative or unknown [7,14,20].

The phiCDKH01 bacteriophage belongs to the former *Siphoviridae* family of phages and has been induced from the clinical isolate of *C. difficile* [21]. Its genome encodes putative regulators of the lysogeny/lytic decision, which have been identified based on the homology search. A putative major repressor gene (*phiCDKH01_17*) has been annotated as a member of the XRE family [22]. Apart from that, two antirepressor genes have also been annotated (*phiCDKH01_20*, *phiCDKH01_24*) along with genes of another three putative transcriptional regulators (*phiCDKH01_19*, *phiCDKH01_22*, and *phiCDKH01_27*).

In this study, we wanted to obtain some insight into the mechanisms responsible for maintaining the lysogeny state of this bacteriophage. We focused on the putative XRE family repressor and tried to unravel its role in this process. XRE (xenobiotic response element) transcription factors are commonly found in bacteria, archaea, bacteriophages, and plasmids. They are known to regulate metabolic processes and environmental responses [23,24], for example, oxidative stress resistance [25] or biofilm formation [26,27]. Nevertheless, two of the most-studied members of the XRE family are CI and Cro repressors of the bacteriophage lambda [28]. Moreover, the *xre* gene from *Bacillus subtilis* 168 has been shown to encode a helix-turn-helix XRE-family-like protein that serves as a transcriptional regulator responsible for the maintenance of the PBSX defective prophage in the lysogeny state [29].

Upon confirmation of its expression in the lysogenic CD34-Sr strain of *C. difficile*, the *phiCDKH01_17* gene product was tested in vitro for its capability of binding to the putative region responsible for controlling the expression of genes involved in maintaining/breaking the lysogeny state of the phiCDKH01 prophage. Having identified the binding site for this putative repressor within the lysogeny control module of the phage genome, we performed in vivo analysis of its influence on the activity of promoters present in this region. Although several works focusing on the identification of major phage repressor genes have been published [30,31], to our knowledge, this is the first study aimed at unraveling mechanisms responsible for maintenance of the lysogeny state of a *C. difficile* prophage.

## 2. Results

### 2.1. The phiCDKH01_17 Is Expressed in Lysogenic Strain C. difficile CD34/Sr

The lysogenic state of the prototypic lambda phage is maintained by a single transcription factor, CI (lambda repressor) [28]. Following this line of reasoning, we wanted to verify whether the *phiCDKH01_17* gene is expressed in the lysogenic strain *C. difficile* CD34-Sr. The presence of the *phiCDKH01_17* transcript was confirmed in the mid-log growth phase cells of the tested strain using RT-PCR reactions (Figure 1) as visualized on the agarose gel by the presence of the band with the expected size of 178 bp.

### 2.2. Cloning and Purification of the phiCDKH01 XRE Protein

Upon verification of *phiCDKH01_17* expression, we cloned this gene into the pBAD expression vector [32] along with the His-tag encoding sequence at its 3′-end to enable further purification of the overexpressed protein using affinity chromatography. The purified protein could be visualized with the use of SDS-PAGE as a single band with a molecular mass corresponding to approximately 20 kDa, which agreed with the calculated molecular mass of the His-tagged XRE protein (21.7 kDa) (Figure 2).

### 2.3. The XRE Protein Binds to the Putative Lysis/Lysogeny Regulatory Region within the phiCDKH01 Genome

A region of the phiCDKH01 genome encompassing genes *phiCDKH01_17* (XRE family transcriptional regulator), *phiCDKH01_20* (Ant antirepressor), and *phiCDKH01_24* (KilAC antirepressor) seems to be particularly interesting in light of the presumed role in regulation of the maintenance of the lysogeny state of the prophage. With the regard to the organization of such regions in the genomes of other bacteriophages, we wanted to check whether any putative promoters could be located within that genome module. A scan with PhagePromoter [33] enabled us to map within this region several theoretic phage promoters that passed a 90% threshold of chances for being active ones. Interestingly, two such promoters were mapped directly upstream of the XRE-encoding gene and one promoter upstream of the gene coding for the putative Ant antirepressor (Figure 3).

Based on these results, we selected three regions that were used to perform electrophoretic mobility shift assays (EMSA) with the purified XRE protein. Migration of all three DNA fragments was retarded in the gel, suggesting the binding of the XRE protein (Figure 4).

At the same time, migration of the control DNA fragment (a fragment of 16S rDNA of *E. coli*) was not affected by the presence of XRE (Figure 4 NC), suggesting specificity of binding to selected regions of the phiCDKH01 genome. In the series of EMSA assays, we were able to narrow down to 30 nt the length of the S2 DNA fragment retaining its capability of binding the XRE protein. The minimal XRE-binding sequence, as indicated with the box in Figure 2, overlaps 6 nt of one of the putative promoters located upstream of the XRE-encoding gene (*phiCDKH01_17*) and lays 6 nt upstream of the putative promoter likely to drive transcription of antirepressor genes. In spite of identification of the XRE-binding site, we were unable to propose its possible consensus sequence (Figure 5).

### 2.4. Activity of Promoters Located within the Putative Lysis/Lysogeny Regulatory Region Are Controlled by the XRE Protein

In the case of lambda phage, the CI repressor regulates the activity of a promoter responsible for the transcription of its own gene as well as the activity of a promoter responsible for expression of the Cro repressor, which counteracts the action of the CI [28]. Having confirmed in vitro binding of the XRE protein to the putative lysis/lysogeny regulatory region, we wanted to ask whether this protein indeed regulates the activity of putative promoters located within it. Since consensus sequences of predicted phiCDKH01 promoters match the consensus sequence binding of the *E. coli* housekeeping sigma RNA polymerase subunit, it was very likely that such promoters could be functional in these bacteria. Therefore, we decided to construct a plasmid harboring the putative lysis/lysogeny regulatory region along with two genes encoding fluorescent proteins, enabling monitoring of the activity of putative promoters. The XRE-encoding gene (*phiCDKH01_17*) was replaced with the gene coding for the GFP protein. The gene coding for the Crimson fluorescent protein was cloned in place of the *phiCDKH01_19* gene, which directly precedes the Ant antirepressor encoding gene in the phiCDKH01 genome. Since both proteins significantly differ in the excitation/emission characteristics (GFP: 490/509 nm, Crimson: 611/646 nm), we were able to simultaneously monitor their fluorescence. The obtained plasmid was named pMR01 (Figure 6).

In the second construct, named pMR02, we added to pMR01 the XRE-encoding gene (*phiCDKH01_17*) along with the arabinose-inducible promoter and araC gene of the pBAD expression vector origin to enable the induction of XRE expression (Figure 6). In the case of *E. coli* strains transformed with both plasmids, we were able to observe fluorescence of the GFP and Crimson proteins, which proved that the promoters present in the cloned region of the phiCDKH01 genome were active. In the case of the strain harboring the pMR01 plasmid, we observed approximately constant GFP and gradually increasing Crimson signals (Figure 7).

This reflects the activity of the promoters driving the transcription of the XRE-encoding gene (*phiCDKH01_17*) and the activity of unrepressed promoter responsible for the transcription of genes coding for antiterminators (*phiCDKH01_20* and *phiCDKH01_24*). In the case of the uninduced culture of the strain harboring the pMR02 plasmid, the overall fluorescence patterns were similar to those observed for the strain with the pMR01 plasmid. Upon the addition of arabinose, we observed a slight decrease in the GFP signal (Figure 7), suggesting a partial repression of promoters responsible for the expression of the XRE-encoding gene. The Crimson signal, on the other hand, exhibited no increase (Figure 7), suggesting complete repression of the promoter of genes encoding the antirepressors.

A two-way ANOVA was performed to analyze the statistical significance of the differences observed for each strain and time point. Appendix A contains a breakdown of the significance and nonsignificance of the associated *p*-values. In the case of the GFP signal, there was not a statistically significant interaction between the effects of time and strain (*p* = 0.357). A simple main effects analysis showed that the differences in the GFP fluorescence were statistically significant (*p* < 0.001) only when compared between different strains. This confirms the observations described above, that the differences in the GFP fluorescence observed for each strain did not significantly differ between the analyzed time points.

In the case of the Crimson signal, we observed a statistically significant interaction between the time and strain (*p* < 0.001). Also, a simple main effects analysis revealed that both the time and strain had a statistically significant effect on the Crimson fluorescence (*p* < 0.001). The results of this analysis confirm the observation that the significance of the differences in the recorded Crimson fluorescence signals at each time point depends on the investigated strain.

## 3. Discussion

*Clostridioides difficile* is an important human pathogen causing an estimated half a million infections in the United States each year [34]. The virulence of *C. difficile* is strictly associated with the production of toxins by these bacteria [4]. It is noteworthy to mention that toxin production can by modified by lysogenic bacteriophages present in their genomes [12].

The temperate nature of all so far known *C. difficile* phages raises an intriguing question about the mechanisms controlling the maintenance of the lysogeny state. In the current literature, there are no reports directly targeting this issue in the context of these bacteria. With the growing number of available sequenced genomes of both *C. difficile* bacteriophages and bacterial strains, the homology searches enable quick identification of possible lysogenic hosts. Nevertheless, the sole fact of the presence of genes encoding putative proteins involved in the control of the lysis/lysogeny decision and/or the maintenance of the prophage in the host genome is not enough to propose a mechanism regulating a particular phage life cycle.

The phiCDKH01 phage encodes several putative transcriptional regulators [21]. In this work, we focused on the *phiCDKH01_17* gene encoding an XRE family protein. Proteins of this family are encoded by various bacteriophages and their role, among others, is to control the state of lysogeny. The best-studied examples of such proteins are two major repressors of the prototypical temperate phage lambda. The CI repressor prevents the onset of the lytic development while the Cro repressor counteracts its action inhibiting the lysogeny [19]. Genes encoding homologs of both repressors can be found in genomes of different *C. difficile* phages; nevertheless, this observation is not universal for all bacteriophages infecting these bacteria. A well-documented example of such bacteriophage is the phiCD38-2 phage, which harbors a gene encoding only the CI homolog [9]. Other *C. difficile* phages, such as phiC2, phiMMP02, and phiCD27 do not encode identifiable homologs of both CI and Cro [7,14,20]. An interesting example of the XRE family protein is the XRE encoded by the defective PBSX prophage present in the genome of the *Bacillus subtilis* 168. Deletion of the gene encoding this protein is lethal to these bacteria since it directly leads to the induction of the prophage and resulting lysis of the host cell [29]. The phiCDKH01 genome contains two annotated genes encoding putative antirepressors located in the proximity of the *phiCDKH01_17* gene (XRE regulator). Interestingly, both putative proteins share a 79% sequence identity and contain a phage antirepressor KilAC domain (CDD database: cl01462), which is characteristic of phage antirepressors counteracting the activity of major phage repressors [35,36]. Because of that, this region of the phiCDKH01 genome is likely to be important for establishing and maintaining the lysogeny. Putative phage promoters P1–P4 mapped in this region strengthen this hypothesis (Figure 3), especially in connection with the observed in vivo activity (Figure 7). Binding of the phiCDKH01 XRE protein to this putative regulatory region (Figure 3) evidently inhibits the activity of the P3 promoter, supposedly driving the transcription of antirepressor genes.

One would expect that the fluorescent signals of the GFP and Crimson observed for both strains ERM01 and ERM02 should be the same in the case of uninduced cultures since, in such a situation, no XRE is expected to be present in the cells. While EMR01 simply lacks the XRE-encoding gene, the absence of arabinose should prevent the *p*_BAD_ promoter from driving transcription of the *phiCDKH01_17* (*XRE*) gene in the cells of the ERM02 strain. On the other hand, in the case of cultures induced with arabinose, we would expect no differences in the signal of both fluorescent proteins in ERM01. In ERM02, a possible decrease in GFP fluorescence is possible due to a partial inhibition of the phage promoter/promoters driving the transcription of the XRE-encoding gene caused by the growing amount of XRE in the cell. A similar situation is observed in phage lambda, where high amounts of the CI repressor cause inhibition of the transcription of its own gene driven from the *p*_RM_ promoter [28]. Simultaneously, the fluorescence observed for the Crimson protein in ERM02 should exhibit no increase due to the repressor activity of XRE toward the P3 promoter. Meanwhile, in the case of uninduced ERM02 cultures, we observed A higher GFP fluorescence in comparison to ERM01 and almost no further increase in the Crimson signal upon a 2 h time point visible in ERM01. Such a result can be explained by the leaking of the *p*_BAD_ promoter, which in turn results in a slow accumulation of the XRE protein in uninduced ERM02 cells. Also, here an analogy to the phage lambda can be made since the CI protein at moderate concentrations activates transcription of its own gene [28]. The phiCDKH01 XRE protein might activate promoter/promoters responsible for the transcription of the *phiCDKH01_17* (*XRE*) gene and partially inhibit the P3 promoter. The lower Crimson signal observed in the case of the induced ERM01 strain might reflect the influence of some intrinsic factors becoming active in the presence of arabinose.

Our results indicate that the *phiCDKH01_17* (*XRE*) gene is expressed in the cells of the phiCDKH01 host (CD34/Sr clinical isolate of *C. difficile*) (Figure 1). For most of the lysogenic strains, only a few phage genes are expressed, with one of them encoding a major phage repressor responsible for the maintenance of the lysogeny state [37,38]. Results of global transcriptome analysis of a *C. difficile* strain carrying the phiCD38-2 prophage confirm these observations. A moderate expression of the putative CI repressor of this bacteriophage (ϕCD38-2_gp39) is sufficient to maintain the lysogeny [30].

We have shown that the XRE protein binds to three fragments of the analyzed region of the phiCDKH01 genome (Figure 4). Interestingly, two of these regions, S1 and S3, do not overlap. Multiple sequence alignment of these DNA fragments led us to the conclusion that this protein binds to the A/T-rich sequences (Figure 5); nevertheless, we were unable to suggest a possible consensus sequence. Although such sequences were proposed for the mentioned XRE-family proteins (lambda CI and Cro [39], *B. subtilis* XRE [40]), they seem to exhibit no similarity to the binding sequence of the phiCDKH01 XRE protein. It is also important to mention that phage-encoded XRE-family proteins are capable of binding to the host chromosome and influencing the bacterial defense mechanisms and viral fitness [41]. The lack of an identified consensus sequence for the phiCDKH01 XRE-binding site makes difficult speculations on such influence regarding the host of this bacteriophage.

To conclude, the XRE protein encoded by the *phiCDKH01_17* (*xre*) gene is capable of binding to the putative regulatory region responsible for the maintenance of the phiCDKH01 phage lysogeny. This binding represses the activity of promoters responsible for the transcription of annotated antirepressor genes. With regard to the literature data, our results suggest the phiCDKH01 XRE protein to be the major repressor of this bacteriophage. An unanswered question remains as to how this protein influences transcription of other phiCDKH01 genes, whose products are involved in the lytic development, opening the way to further research on the regulation of lifestyles of this bacteriophage.

## 4. Materials and Methods

### 4.1. The phiCDKH01_17 Encodes a Putative XRE-Family Transcription Factor

The *phiCDKH01_17* open reading frame of the phiCDKH01 bacteriophage has originally been annotated as a gene encoding an XRE family transcriptional regulator [21]. We used the amino acid sequence of this protein to perform a homology search using the NCBI BLAST suite [42]. The results of this search indicated that this protein contains two helix-turn-helix domains characteristic of XRE-family-like proteins (CDD database: cd00093). The first 100 hits returned by the NCBI BLAST suite were XRE family transcriptional regulators from various *C. difficile* strains exhibiting a sequence identity to phiCDKH01_17 of over 95% (30 hits), between 75 and 95% (25 hits), between 51 and 75% (41 hits), and between 48 and 51% (4 hits) (Appendix A). Therefore, in this study, we refer to the phiCDKH01_17 as the XRE protein.

### 4.2. Bacterial Strains and Culture Conditions

*E. coli* strains (Table 1) were grown with an agitation (180 rpm, 37 °C) in LB (Miller) medium (Merck, Darmstadt, Germany) containing ampicillin (100 μg/mL) unless indicated otherwise. The *C. difficile* strain (Table 1) was grown as a stationary culture in BHI broth (Merck, Darmstadt, Germany) at 37 °C under anaerobic conditions.

### 4.3. RNA Isolation from C. difficile, Reverse Transcription and Analysis of the phiCDKH01_17 Gene Transcription

The total RNA was prepared from 5 mL of log phase BHI broth cultures of *C. difficile* 34-Sr strain using the SV Total RNA Isolation System (Promega, Wisconsin, MI, USA). The isolated RNA was treated with DNase I to remove genomic DNA and reverse transcription was performed using the PrimeScript™ RT Reagent Kit with gDNA Eraser (TaKaRa, Saint-Germain-en-Laye, France). The absence of contaminating genomic DNA was verified by performing a 35-cycle PCR in the presence of 200 ng total RNA.

A sample of cDNA was used as a template in a PCR reaction with XRE-RT-up and XRE-RT-dn primers and DreamTaq Green PCR Master Mix (ThermoFisher Scientific, Waltham, MA, USA). The obtained products were visualized using agarose gel electrophoresis.

### 4.4. Phage Induction, Purification and DNA Extraction

A total of 500 mL of log phase BHI broth culture of the *C. difficile* 34-Sr strain was exposed to mitomycin C (Abcam, Cambridge, UK) at a final concentration of 3 µL/mL for 12 h. Lysate was centrifuged at 3400× *g* for 30 min and filtered through a 0.45 μm filter membrane (Merck, Darmstadt, Germany). Phage lysate was concentrated using polyethylene glycol (PEG) precipitation as described in [44] with some modifications. Briefly, 10% PEG-8000/1 M NaCl solution was added to the supernatant fraction and incubated overnight. The sample was pelleted down for 30 min at 12,000× *g* and the pellet was dissolved in an SM buffer (150 mM NaCl, Tris-HCl pH 6.5, 10 mM MgCl_2_, 1 mM CaCl_2_). Next, an equal volume of chloroform was added to the phage solution and the mixture was centrifuged at 13,000× *g* for 5 min to induce phase transition. The upper phase containing phages was collected and stored at 4 °C until use.

The DNA of the isolated phage was extracted using a modified phenol/chloroform method as described elsewhere [45]. Briefly, 3 mL of concentrated phages were treated with 50 μg/mL DNaseI and 50 μg/mL RNaseA (Merck, Darmstadt, Germany) for 1 h at 37 °C. After incubation of 10 μL of proteinase K (20 mg/mL) was added and incubated at 37 °C for 10 min to remove the DNAse I and RNAse A, the reaction was stopped by adding 0.5 volume of an SDS-Mix (0.5 M Tris-HCl pH 9.0, 0.25 M EDTA, 2.5% SDS) and incubating at 65 °C for 30 min. Next, 0.125 mL of 8 M potassium acetate was added, mixed well, incubated on ice for 30 min, and centrifuged at 13,000× *g* for 10 min at 4 °C. An equal volume of 25:24:1 phenol–chloroform–isoamyl alcohol (Merck, Darmstadt, Germany) was added to the solution. The mixture was then vigorously mixed and centrifuged at 13,000× *g* for 15 min at room temperature. The aqueous phase was collected. The phenol–chloroform–isoamyl alcohol extraction step was repeated twice. Phage genomic DNA was precipitated with isopropanol. The pellet was resuspended in 50 μL of 10 mM Tris–HCl pH 8 and stored at −20 °C until use.

### 4.5. Plasmid Construction

#### 4.5.1. phiCDKH01

The plasmid for overexpression of the phiCDKH01 XRE protein was prepared using a pBAD expression vector (ThermoFisher Scientific). A phiCDKH01 genome fragment was amplified with Q5 high-fidelity DNA polymerase (New England Biolabs, Ipswich, MA, USA) using XRE-his-up and XRE-his-dn primers and phage genomic DNA as template. An amplified fragment and vector DNA were digested with NheI and EcoRI restriction enzymes (ThermoFisher Scientific, Waltham, MA, USA), purified, and ligated using T4 DNA ligase (ThermoFisher Scientific, Waltham, MA, USA). The ligation mixture was used for the transformation of *E. coli* DH5α. Ampicillin-resistant colonies obtained after overnight incubation were verified with PCR for the presence of cloned insert and selected for further analysis. The constructed plasmid was named pBAD-XRE.

Plasmids for analyses of the transcriptional activity of putative phage promoters, as well as for testing the influence of the phiCDKH01 XRE protein (Figure 6), were constructed using the NEBuilder^®^ HiFi DNA Assembly Master Mix (New England Biolabs, Ipswitch, MA, USA).

#### 4.5.2. pMR01

The pMR01-plasmid-harbouring putative promoters of the *phiCDKH01_17* gene (XRE) and genes encoding annotated antirepressors (*phiCDKH01_20* and *phiCDKH01_24*) along with genes encoding the GFP and Crimson proteins were assembled using the following fragments: a phiCDKH01 genome region of interest encoding putative promoters amplified with XRE_reg_fwd and XRE_reg_rev primers and phage genomic DNA as a template; a GFP-encoding fragment amplified with GFP_for and GFP_rev primers and pBAD plasmid DNA as a template; a fragment-encoding Crimson protein amplified with Crimson_for and Crimson_rev primers; and a pE2-Crimson vector DNA (TaKaRa, Saint-Germain-en-Laye, France) as a template. A HincII-linearized pUC19 was used as a vector for construction. Assembly reaction and following transformation of *E. coli* DH5 α were performed following the manufacturer’s protocol.

#### 4.5.3. pMR02

Construction of the pMR02 plasmid was based on the pMR01 construct. An EcoRI-linearised pMR01 plasmid was assembled with a pBAD-XRE fragment containing the *araC* gene encoding transcriptional regulator along with its promoter, *phiCDKH01_17* (XRE) encoding gene under the control of a pBAD promoter, followed by a transcriptional terminator. The pBAD-XRE fragment was amplified using XRE-pBAD_fwd and XRE-pBAD_rev primers and the pBAD-XRE plasmid DNA as a template. The assembly and transformation of *E. coli* DH5α were performed following the manufacturer’s protocol.

Primers used for amplification of the DNA fragments used for constructions are listed in Appendix A. The constructs prepared with the NEBuider^®^ HiFi DNA Master Mix were designed in the NEBuilder^®^ Assembly Tool available on the website of New England Biolabs.

### 4.6. Overexpression and Purification of phiCDKH01 XRE Protein

The overnight culture of a strain harboring pBAD-XRE plasmid was refreshed by a 1:100 dilution in a fresh LB medium. Bacteria were grown with agitation at 37 °C until the culture reached an OD_600_ of approximately 1. To induce the expression of the phiCDKH01 XRE protein, arabinose was added to the final concentration of 0.05% and culturing was continued for another 2 h. Next, bacteria were pelleted by centrifugation and batch purification was performed with the use of an Ni-NTA resin (Qiagen, Hilden, Germany) following the manufacturer’s protocol. The purified protein was dialyzed against a PBS buffer (137 mM NaCl, 2.7 mM KCl, 10 mM Na_2_HPO_4_, 1.8 mM KH_2_PO_4_), supplemented with glycerol to a final concentration of 30%, and stored at −20 °C.

### 4.7. Gel Retardation Electrophoresis

Assays were carried out using one of the double-stranded DNA probes. Probes S1, S2, and S4 were amplified in PCR reactions with appropriate primers (Appendix A) and the phiCDKH01 phage genomic DNA as a template. The negative control probe NC was amplified in PCR reactions with 16S-Ec-up and 16S-Ec-dn primers and *E. coli* DH5α genomic DNA as template. The probe BIND was obtained by hybridization of XRE-BIND-up and XRE-BIND-dn oligonucleotides. A total of 0.1 pmoles of the DNA probe were incubated with a fixed amount of 0–200 ng of purified phiCDKH01 XRE protein in 50 mM Tris-HCl pH 7.5 containing 50 mM NaCl, 200 mM KCl, 50 mM MgCl_2_, 5 mM EDTA, 5 mM DTT 0.25 mg/mL BSA for 15 min at room temperature. Upon incubation, the samples were mixed with TriTrack DNA Loading Dye (Thermo Fisher Scientific), loaded onto the 5% TBE polyacrylamide gel, and run at a constant voltage of 10 V/cm for 1.5 h. The gel was stained in 0.5 μg/mL solution of ethidium bromide and visualized using the ChemiDoc Imaging System (Bio-Rad, Hercules, CA, USA).

### 4.8. Monitoring Putative Phage Promoters Activity

Overnight cultures of the bacteria harboring the plasmids pMR01, pMR02, and pBAD-XRE were refreshed by a 1:100 dilution in a fresh LB medium and incubated with agitation at 37 °C until they reached an OD_600_ of approximately 1. At that time, the cultures were split into two aliquots. In the case of each strain, one aliquot was induced by adding arabinose to a final concentration of 0.05%. All cultures were incubated with agitation at 37 °C and the samples were collected every 60 min for the next 4 h. The 100 μL samples were transferred into the 96-well transparent plate and an optical density at 600 nm was measured using a Synergy H1 microplate reader (BioTek, Winooski, VT, USA). Another 100 μL of samples were transferred into the 96-well black plate and the fluorescence of the GFP (488 nm excitation/525 nm emission) and Crimson (611 nm excitation/646 nm emission) proteins was measured using the same device.

### 4.9. Statistical Analysis

Data were analyzed with two-way ANOVA using an R package 4.4.0 (https://www.R-project.org/). Prior to analysis, assumptions of the test were verified. The Tukey Honestly Significant Differences test was used as in post hoc analyses.

## Figures and Tables

**Figure 1 ijms-25-05662-f001:**
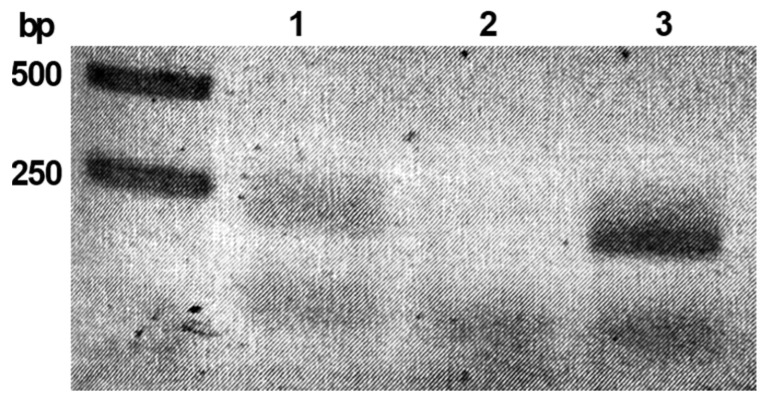
Expression of the *phiCDKH01_17* gene in *C. difficile* CD34-Sr clinical isolate. PCR products of reactions with primers specific for *phiCDKH01_17* were visualized with agarose gel electrophoresis. Reactions were performed with the following templates: CD34-Sr cDNA (lane 1), CD34-Sr total RNA (lane 2), CD34-Sr genomic DNA (lane 3). Bands on the marker lane are labeled with corresponding DNA fragment lengths.

**Figure 2 ijms-25-05662-f002:**
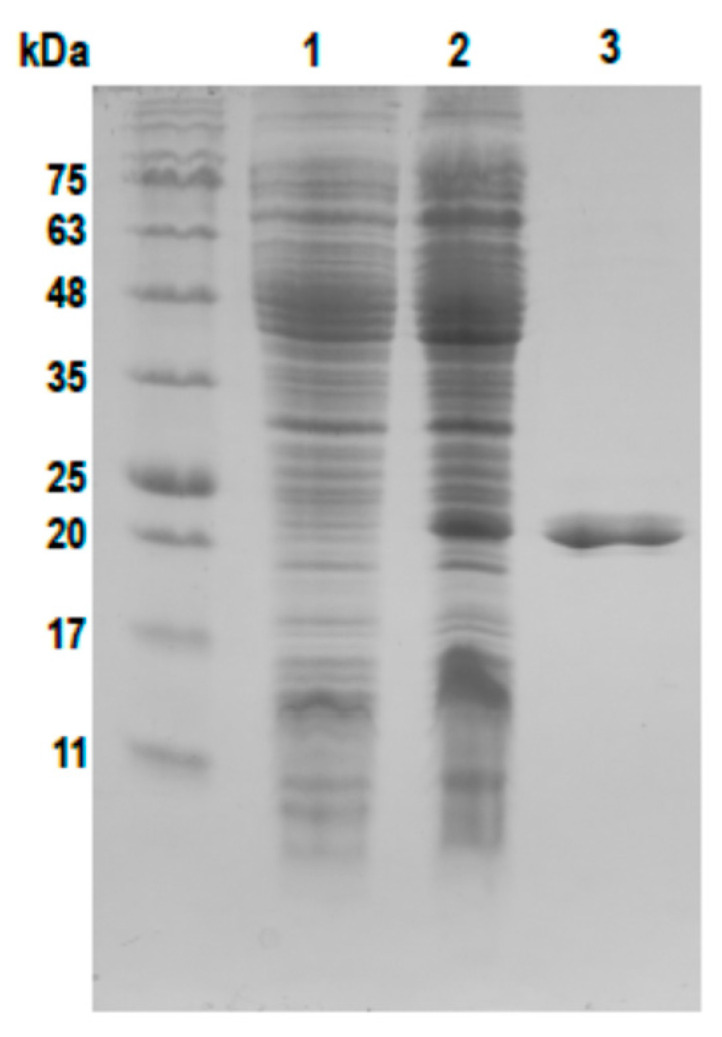
Overproduction and purification of phiCDKH01 XRE protein. Total protein was extracted from EMB2401 strain culture uninduced (lane 1) and with induced lane 2) with 0.05% arabinose. Lane 3—purified protein phiCDKH01 XRE.

**Figure 3 ijms-25-05662-f003:**
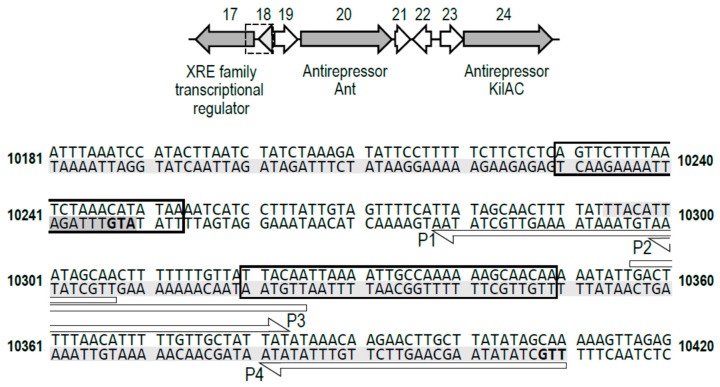
Organization of the phiCDKH01 genome region encoding genes putatively involved in regulation of the lysis/lysogeny decision. Numbers above the schematic drawing of gene layout indicate annotated open reading frames. Genomic sequence corresponds to the region marked on the schematic drawing with dashed line box. Shaded parts of the genome sequence correspond to open reading frames originating with START codons marked in bold. Open arrows labeled with P1, P2, P3, and P4 reflect positions of putative phage promoters. The boxed sequences correspond to the XRE protein binding sites.

**Figure 4 ijms-25-05662-f004:**
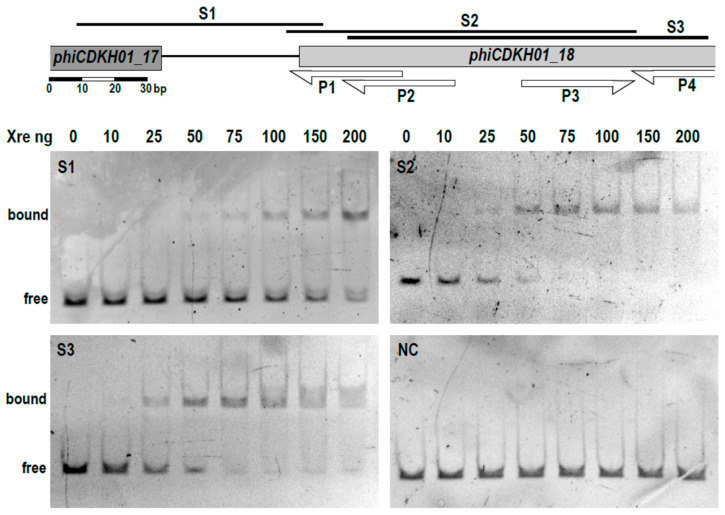
The XRE protein binds to the putative lysis/lysogeny decision region of the phiCDKH01 phage. DNA binding of XRE was tested using EMSA assay. The schematic drawing visualizes phiCDKH01 genome fragments (**S1**–**S3**) used in DNA-binding assays and their positions corresponding to the putative phage promoters (P1, P2, P3, P4). The XRE binding with DNA fragments is observed as retardation of their electrophoretic mobility in the polyacrylamide gels upon staining with ethidium bromide. (**NC**)—products of EMSA assay with 16S rDNA fragment of *E. coli* DH5α used as a negative control.

**Figure 5 ijms-25-05662-f005:**
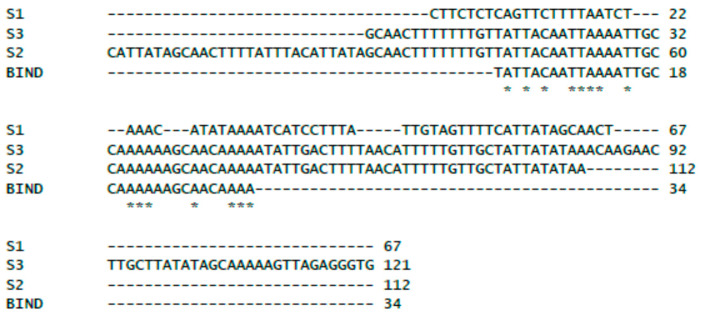
Multiple sequence alignment of XRE-binding fragments of phiCDKH01 genome. The sequence marked as BIND corresponds to the minimal XRE-binding DNA fragment. Asterisks indicate bases identical in all aligned sequences.

**Figure 6 ijms-25-05662-f006:**
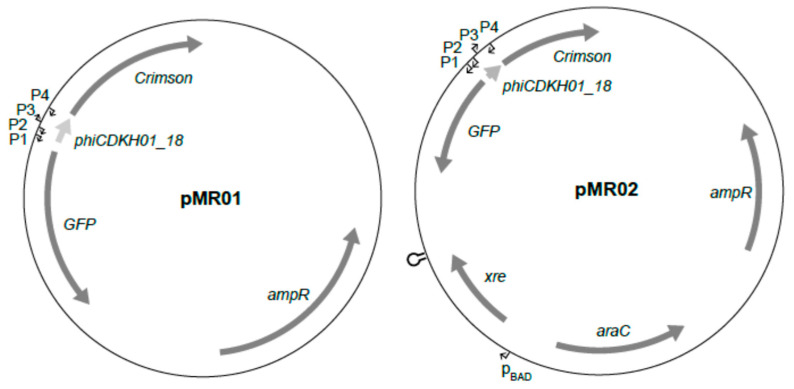
Schematic drawing of plasmid constructs used for in vivo testing of the XRE activity toward the putative phage promoters. Putative promoters and *p*_BAD_ promoter are indicated with black arrows. A hairpin symbol represents a transcriptional terminator.

**Figure 7 ijms-25-05662-f007:**
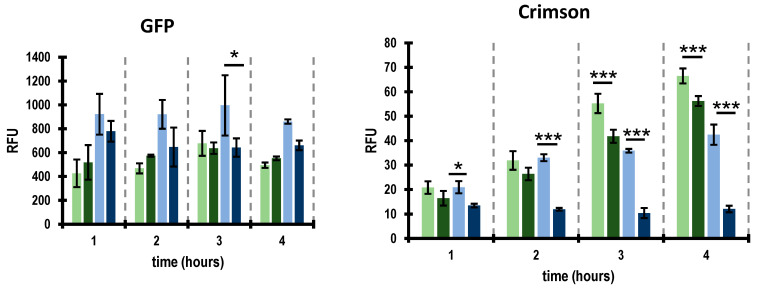
Activity of promoters present in putative lysis/lysogeny region of the phiCDKH01 genome as assessed by fluorescence of GFP and Crimson proteins. Activities of putative promoters driving transcription of annotated XRE regulator and antirepressor genes correspond to fluorescence of GFP and Crimson, respectively. Assayed strains were harboring the following plasmids: pMR01 (strain EMR01, green bars) and pMR02 (strain EMR02, blue bars). Activities of promoters were tested in uninduced (light green and light blue bars) and induced (dark green and dark blue bars) cultures with 0.05% arabinose. Relative fluorescence units (RFU) represent means of three repeats (*n* = 3) of the experiment. Error bars indicate standard deviations. * *p* < 0.05, *** *p* < 0.001.

**Table 1 ijms-25-05662-t001:** Strains used in the study.

Name	Relevant Genotype	Source of Reference
*Escherichia coli*		
DH5α	*fhuA2 lac(del)U169 phoA glnV44 W809 lacZ(del)M15 gyrA96 recA1 relA1endA1 thi-1 hsdR17*	[43]
EMB2401	DH5α pBAD-XRE	This study
EMR01	DH5α pMR01	This study
EMR02	DH5α pMR02	This study
*Clostridioides difficile*		
CD34-Sr		[21]
phiCDKH01 phage		[21]

## Data Availability

Data is contained within the article and Appendix A.

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
