# Peer review of "Insight into the Mechanism of Lysogeny Control of phiCDKH01 Bacteriophage Infecting Clinical Isolate of Clostridioides difficile"

_ijms, 2024, doi:10.3390/ijms25115662_

Round 1

Reviewer 1 Report

Comments and Suggestions for Authors

The article "Insight into the mechanism of lysogeny control of phiCDKH01 bacteriophage infecting clinical isolate of Clostridioides dificile" describes the identification of promoter region within the phage phiCDKH01 genome bound by the newly identified Xre repressor protein. The authors describe molecular techniques to identify the region of the promoters where Xre binds. The overall manuscript is good, but there are some changes necessary to clarify the experiment used to demonstrate the effects of Xre binding on the expression from both the xre promoter as well as the anti-repressor promoter (ant). My suggestions are below.

Major comments-

1. In Figure 1, please provide information in the text as to the expected size of the PCR fragment. The gel visually confirms expression, but to readers who are not as based in molecular methods, the presence of the lower band could be confusing. 

2. The experimental design represented by Figure 6 is quite clever. That said, it needs to be conveyed differently as it is confusing in its current state. In the text, it would be helpful to have discussion of what the expected results would be for each strain under non-induced and induced conditions so that the reader can track what is seen in the graphs. Along these lines, my understanding is that the black bars are strains that should be expressing inducible Xre, the gray bars have GFP expressed from the xre promoter and Crimson expressed from the anti-repressor promoter, and the white bars should have the fluorescent proteins expressed from the promoters as well as inducible Xre. Thus, black bars in A and B are showing background levels of fluorescence in the GFP channel correct? These bars should not really increase or decrease with induction as its cellular autofluorescence that's being measured. 

Continuing this, the gray bars shouldn't show change with addition of arabinose either as they aren't inducible and there's nothing repressing them. However, it looks like there is a decrease that occurs in promoter expression with arabinose (specifically with Crimson at 3 hrs). Finally, the white bars should show decreases in expression from the anti-repressor promoter with induction as Xre binds this promoter region. It also binds and represses its own promoter region although less strongly correct?

So, all of that needs to be spelled out better in the text or a pictorial of what is occurring should be included as an addition to this figure. 

Additionally, the method of labeling significance doesn't work. May I suggest incorporating a table with a breakdown of significance and non-significance with associated p-values to clearly demonstrate what the differences are. 

3. Plasmid maps should be included.

Minor suggestions:

1. Change line 13 to "So far, all known bacteriophages..."

2. In line 44, change to "It is worth mentioning...."

3. In line 50, change to "... leading to DNA damage...."

4. in line 53, change to ".... keeping and breaking of the lysogeny state..."

5. In line 57, get rid of the "s" on details.

6. In line 90, change to ".... first study aimed at unraveling...."

7. In line 136, define EMSA

8. in line 138, change to "... suggesting binding of the Xre protein."

9. In line 180, italicize "E. coli"

10. Change "rises" to "raises" in line 224

11. in line 344, you could change Invitrogen to ThermoFisher as they are now the same

12. In line 369, did you mean to say linearised pMR01?

Comments on the Quality of English Language

The English language in this document is fine. There are a few necessary edits, but are easily made.

Reviewer 2 Report

Comments and Suggestions for Authors

The authors present a study concerning the analysis of a lambda-like lysogeny-decision region in a C. diff phage. The results are mostly convincing and worthy of publication.

However, several issues need to be cleared up beforehand:

1)      It is a bit confusing  how the minimal repressor binding region is defined – from figure 4, it looks like the repressor shows electrophoretic shifts of all three segments tested. Yet the minimal binding area as defined in figure 1 does not cover the region S1 that according to Figure 4 is still being bound. Figure 5 shows the binding area of S1 – so it should be indicated in Figure 2. The text also says it was impossible to determine a consensus sequence, but does figure 5 not show exactly that (as BIND)?

2)      Results in Figure 6A/B seem muddled by autofluorescence and possibly transcriptional readthrough from the pBAD promoter, which probably lead to the lack of statistical significance discussed in lines 203-210. In 6A and B, pMR01 (carrying gfp under control of the putative promoter) shows the same/very similar fluorescence intensity as pbad-XRE, which does not carry gfp -  so we are probably observing autofluorescence at the GFP-wavelength. However, there is considerably higher fluorescence for pMR02, especially in the uninduced state – normally, we would expect reduced fluorescence because the leaky promoter pBAD would still produce an amount of XRE to reduce GFP. Yet we are seeing INCREASED GFP fluorescence – is it possible that there is some degree of transcriptional readthrough despite the terminator sequence that ends up causing higher GFP expression? No plasmid map is available for the reviewer to quickly rule out this possibility. I would suggest repeating these experiments with crimson instead of gfp, and perhaps re-orienting the pBAD promoter/having it on a different plasmid entirely.

In contrast, the evidence for control of phiCDKH01_19 is convincing and the data just as we would expect – no background for pbad-xre, increasing fluorescence for pMR01 over time, decreasing for pMR02 over time despite lack of induction because of leaky promoter. Then clear reduction in fluorescence under inducing conditions, unlike for gfp/ phiCDKH01_17

It does not help that Figure 6 is very hard to interpret in how it is laid out. Why are induced and uninduced samples not right next to each other? Why all these letters instead of just showing corresponding samples next to each other with stars where differences are significant? To compare I have to draw bars with my mouse pointer and jump between different figures, not great.

Further minor comments:

14: “ host cellS they can integrate INTO”

103: Slight confusion on my part – RNA in lane 2 of the figure is non-reverse transcribed, correct? The preceding sentences implies it is all from RT-PCR.

371: pBAD or pBAC promoter?

Round 2

Reviewer 2 Report

Comments and Suggestions for Authors

I still don’t understand why there is only one xre binding site (for S2/S3) indicated in figure 3. Why not a second box for S1/S2, which clearly also binds xre?

See my illustration in the attached word file.

Figure 7 (formerly 6) is now much more clear, and the addition of plasmid maps helps with understanding. The authors explanation for why they see more crimson in the uninduced repressor plasmid than in plasmids with no repressor at all (line 277) is confusing – “presence of moderate amounts of [the inhibitor] activates promoter/promoters responsible for transcription of the xre gene and partially inhibits p3 promoter – this seems like a statement that would either require a citation or indication that this is just guesswork. A similar explanation (in line 280) for why there is less crimson when arabinose is added into a system that has no arabinose promoter is more cautious and makes more sense – line 277 should be similar.

Once these minor issues are clarified the paper is ready for publication.

Author Response

Response:

Figure 1 was modified following Reviewers suggestions. We added another box indicating Xre binding site deduced from sequence alignment of EMSA fragments shown in Figure 5.

We changed discussion of GFP signals observed in EMR02 strains (l.272-274 and 280-282). We referred to lambda phage CI and provided appropriate citation.